# Anti-Insect Properties of *Penicillium* Secondary Metabolites

**DOI:** 10.3390/microorganisms11051302

**Published:** 2023-05-16

**Authors:** Rosario Nicoletti, Anna Andolfi, Andrea Becchimanzi, Maria Michela Salvatore

**Affiliations:** 1Council for Agricultural Research and Economics, Research Center for Olive, Fruit and Citrus Crops, 81100 Caserta, Italy; rosario.nicoletti@crea.gov.it; 2Department of Agricultural Sciences, University of Naples Federico II, 80055 Portici, Italy; 3Department of Chemical Sciences, University of Naples Federico II, 80126 Naples, Italy; andolfi@unina.it (A.A.); mariamichela.salvatore@unina.it (M.M.S.); 4BAT Center-Interuniversity Center for Studies on Bioinspired Agro-Environmental Technology, University of Naples Federico II, 80055 Portici, Italy; 5Institute for Sustainable Plant Protection, National Research Council, 80055 Portici, Italy

**Keywords:** bioactive products, chemodiversity, ecofriendly insecticides, entomopathogenic fungi, insect pest management, symbiotic interactions

## Abstract

In connection with their widespread occurrence in diverse environments and ecosystems, fungi in the genus *Penicillium* are commonly found in association with insects. In addition to some cases possibly implying a mutualistic relationship, this symbiotic interaction has mainly been investigated to verify the entomopathogenic potential in light of its possible exploitation in ecofriendly strategies for pest control. This perspective relies on the assumption that entomopathogenicity is often mediated by fungal products and that *Penicillium* species are renowned producers of bioactive secondary metabolites. Indeed, a remarkable number of new compounds have been identified and characterized from these fungi in past decades, the properties and possible applications of which in insect pest management are reviewed in this paper.

## 1. Introduction

Insects have a tremendous impact on human welfare in terms of both directly affecting our health and damaging crops and foodstuffs. Until recently, the never-ending struggle against these deleterious animals has mainly relied on the use of insecticides; however, many synthetic chemicals proved to have harmful side effects, calling for the introduction of alternative control tools and strategies. Hence, the exploitation of natural sources that could minimize the undesirable outcomes caused by chemicals to health and the environment has become undeniable. In this respect, all sorts of antagonistic organisms have been considered that could exert a natural pressure on pest populations, including entomopathogenic fungi [1]. This category has been defined to comprise fungi able to cause diseases in insects and to spread at epidemic levels in insect populations. These aptitudes characterize a good number of species, such as *Beauveria bassiana*, *Metarhizium anisopliae*, and *Lecanicillium-Akanthomyces* spp., which today are routinely employed in the biological control of several insects [2,3,4]. However, the effects of fungi on insect pests can go beyond the classical pathogenic interactions, involving fine physiological effects that are modulated by secondary metabolites released in the plant tissues [5,6].

Fungi in the genus *Penicillium* have also been reported for their entomopathogenic behavior, and some species have been proposed as effective biocontrol agents, such as *P. corylophilum* and *P. janthinellum*, against mosquitoes [7]. They are among the most prolific producers of compounds displaying bioactive properties with possible applications in human and veterinary medicine, agriculture, and other fields as antibiotics, antitumor drugs, modulators of the immune system, enzyme inhibitors, etc. [8,9,10,11]. This paper presents an overview of the secondary metabolites of these fungi which have displayed various sorts of anti-insect effects and may be considered for possible exploitation in pest management.

## 2. *Penicillium* Species as Insect Associates

*Penicillium* species (Eurotiomycetes, Aspergillaceae) are widespread in every environment on earth, from desert sands to Antarctica to ocean trenches [8,12,13,14]. Their occurrence in ecosystems is even more pervasive as a result of the ability to establish symbiotic relationships with various organisms [11,15,16]. These features of adaptability and eclecticism reflect a notable taxonomic diversity such that the current number of more than 400 species requires being continuously integrated with new findings and descriptions [13].

Until recently, *Penicillium* was used to designate the anamorphic stages of species classified in the genera *Eupenicillium* and *Talaromyces*. After the adoption of the ‘one fungus, one name’ concept in fungal taxonomy, this genus name has come into general use for all species except for those in the subgenus *Biverticillium*; in fact, following accurate phylogenetic studies, the latter have been officially separated and are now classified as *Talaromyces* in the family Trichocomaceae [17]. As a result, biverticillate species, such as *P. funiculosum*, *P. pinophilum*, *P. purpurogenum* and *P. rugulosum*, which are frequently reported as insect associates [18], have not been considered in this review; moreover, species formerly classified in the genus *Eupenicillium* are mentioned according to the revised *Penicillium* nomenclature [17]. While making the recent identifications more reliable, these taxonomic reassessments and the widespread use of DNA sequencing for strain classification have introduced some uncertainty into identifications done in past decades, based exclusively on morphology. Hence, some data collected more than 10 years ago could prove to be incorrect or require revision.

In addition to a series of isolates provisionally identified as *Penicillium* sp., the examination of the available literature indicates that at least 62 *Penicillium* species have been identified in association with insects so far (Table 1).

Most of the findings listed in Table 1 refer to isolations from whole insect bodies, which does not allow for advancing reliable hypotheses about a defined symbiotic role. Rather, in most cases, the occurrence of *Penicillia* can be regarded as deriving from occasional contamination or ingestion. However, some clues about specialization could concern species that were first or exclusively reported as insect associates; this is the case of *P. brocae* [30], *P. mallochii* and *P. guanacastense* [51], *P. costaricense* and the three species *P. camponotum*, *P. infrabuccatum,* and *P. fundyense*, which so far have only found been in the buccal cavities of carpenter ants (*Camponotus* spp.: Hymenoptera, Formicidae) [31]. A mutualistic association has been conjectured in the case of two unidentified species isolated from both leaf rolls and the mycangia of females of the leaf-rolling weevil *Euops lespedezae* (Coleoptera, Attelabidae). In fact, one or the other of these species was isolated from 80% of females, and they proved to be the most dominant fungi in leaf rolls, especially at the egg stage [62]. Later, *P. herquei* was determined to be the dominant species associated with the congeneric *Euops chinensis* [15]. Mutualism has been also considered to possibly characterize the systematic occurrence of several *Penicillium* species in the digestive tracts of kissing bugs (*Triatoma* spp.: Hemiptera, Reduviidae) [25,50]. On the opposite edge of symbiotic relationships, entomopathogenic aptitude has been documented for some common species, such as *P. brevicompactum* [29], *P. citrinum* [39,40], *P. corylophilum*, *P. fellutanum*, *P. janthinellum*, *P. viridicatum,* and *P. waksmanii* [7].

The notable taxonomic diversity of insect-associated *Penicillia* further increases when considering isolations from insect microenvironments or products. For instance, the new species *P. apimei*, *P. fernandesiae*, *P. meliponae,* and *P. mellis* were identified in Brazil from honey, bee pollen or from inside the nests of the stingless bee *Melipona scutellaris* (Hymenoptera, Apidae), along with a series of known and more or less infrequent species, namely *P. brocae*, *P. chermesinum*, *P. citreosulfuratum*, *P. citrinum*, *P. echinulonalgiovense*, *P. fellutanum*, *P. mallochii*, *P. paxilli*, *P. rubens*, *P. sansahense*, *P. sclerotiorum*, *P. shearii*, *P. singorense*, *P. steckii*, *P. sumatrense*, and *P. wotroi* [72]. In addition to bees, the nests of ants, termites, and other social insects have frequently been reported as sources of isolation of *Penicillium* spp. [73]; however, in these cases, a symbiotic connection is not obvious, considering that these fungi are ubiquitous in soil.

## 3. Effects on Insect Viability and Development

The above-considered entomopathogenic aptitude disclosed by some *Penicillium* spp. is thought to be basically related to the capacity by the infectious strains to synthesize bioactive compounds that can help to overcome the defensive barriers provided by the host’s immune system or influence its behavior and fitness [74]. This reasonable hypothesis has stimulated enormous research activity aiming to identify the candidate products that can be credited for these anti-insect properties and can be further investigated for possible application in various fields.

### 3.1. Crude Culture Extracts

Some reports have been limited to preliminary evidence of insecticidal activity by culture extracts, which has not been followed by the characterization of the bioactive compounds. This is the case with dichloromethane extracts from liquid cultures of *P. decumbens* and *P. oxalicum*, displaying insecticidal effects against the large milkweed bug (*Oncopeltus fasciatus*: Hemiptera, Lygaeidae) in assays in which the insects were maintained in Petri dishes containing a deposit of 500 μg cm^−2^ of extract; particularly, extracts from cultures in Wickerham broth of *P. oxalicum* induced 100% mortality [75]. Moreover, as tested for its larvicidal activity on the tobacco cutworm (*Spodoptera litura*: Lepidoptera, Noctuidae) and the southern house mosquito (*Culex quinquefasciatus*: Diptera, Culicidae), ethyl acetate extract from cultures of a *Penicillium* strain related to *P. chrysogenum* induced significant mortality, resulting after dipping of larvae in the extract for 10 s. In greater detail, LC_50_ of 72.205 mg mL^−1^ and 94.701 mg mL^−1^, and LC_90_ of 282.783 mg mL^−1^, and 475.049 mg mL^−1^ were calculated against the two insect species, respectively [76].

Culture filtrates of an unidentified *Penicillium* strain that is endophytic in the leaves of the scrambling shrub *Derris cuneifolia* (=*D. hancei*) caused 100% mortality of second-instar larvae of *S. litura* in a leaf disc feeding assay and 75.10% mortality of the turnip aphid (*Lipaphis erysimi*: Hemiptera, Aphididae) in a soaking assay [77]. Three chromatographic fractions from the chloroform extract of the mycelium of another *Penicillium* strain endophytic in the roots of *Derris elliptica* showed insecticidal effects against *L. erysimi* after 48 h of exposure to a concentration of 1 mg mL^−1^; the induced mortality rates were 57.68%, 63.28%, and 69.74%, respectively [78]. Finally, ethyl acetate extract from the mycelium of an unidentified *Penicillium* strain displayed insecticidal effects against first to fourth-instar larvae of *C. quinquefasciatus* and the yellow fever mosquito (*Aedes aegypti*: Diptera, Culicidae) as assayed at four concentrations (100, 200, 300, and 500 μg mL^−1^); for the two mosquito species, the mortality data were respectively calculated as LC_50_ in the ranges of 7–25.2 μg mL^−1^ and 5.5–6.9 μg mL^−1^, while LC_90_ was in the ranges 12.5–41.7 μg mL^−1^ and 10.4–13.9 μg mL^−1^ [79].

### 3.2. Purified Compounds

#### 3.2.1. Mycotoxins

Mycotoxins have been the subject of enormous investigational activity in light of assessing their effects on a variety of organisms. Insects not only provide a model for toxicological studies but also can pose a threat to human and animal health by acting as a vehicle for contamination of foodstuffs with mycotoxin-producing fungi. On the other hand, direct exposure to mycotoxins is considered a factor possibly affecting the spread of insect harriers in foodstuffs. Some storage fungi attract insects as food sources and promote population increases; others produce metabolites with a repellent effect [80]. Indeed, the mycotoxins-insects interaction is quite variable in its effects and calls into question the basic role played by cytochrome P-450 in mycotoxin detoxification [81]. Of course, *Penicillium* spp. are particularly involved in this biological struggle, with reference to their renowned implications in the contamination of food and feed with mycotoxins, such as citrinin, ochratoxin A, patulin, and more [16].

Mycotoxins (Figure 1) may impact insect physiology in multiple ways. In one of the first studies published on this topic, citrinin was found to be repellent toward the confused flour beetle (*Tribolium confusum*: Coleoptera, Tenebrionidae) when offered in whole wheat flour; nevertheless, development of the insect was generally promoted by the presence of mycotoxin-producing *Penicillia* in the flour [82]. In additional studies performed by the same research group, ochratoxin A, citrinin, rubratoxin B, patulin, penicillic acid, and oxalic acid were fed at various concentrations to *T. confusum*, the cigarette beetle (*Lasioderma serricorne*: Coleoptera, Ptinidae), and the black carpet beetle (*Attagenus megatoma*: Coleoptera, Dermestidae) in whole wheat flour. Penicillic and oxalic acids were not toxic to these insects at any concentrations; ochratoxin A and citrinin inhibited larval growth of *A. megatoma* at 10 and 1000 ppm, respectively, while rubratoxin B had no effect; citrinin and rubratoxin B inhibited larval growth of *T. confusum* and *L. serricorne* at 1000 ppm only; and patulin inhibited growth of *T. confusum* at 1000 ppm, either with or without yeast supplementation in the diet. Moreover, growth of *L. serricorne* larvae was slower when patulin was added to the diet at any concentration, without yeast. Finally, reproduction of *T. confusum* was impaired by citrinin, patulin, and ochratoxin A at the highest concentration tested, while only citrinin affected the reproduction of *L. serricorne* [83]. Further assays were performed by adding the same mycotoxins in 10-fold concentrations to wheat flour administered to the Mediterranean flour moth, *Anagasta* (=*Ephestia*) *kuhniella* (Lepidoptera, Pyralidae), with and without yeast supplementation. Larval growth was inhibited by citrinin, ochratoxin A, patulin, and rubratoxin B; and ochratoxin A (10 ppm), citrinin, and rubratoxin B (100 ppm) caused significant mortality, with no larvae surviving at 1000 ppm citrinin and 100 ppm ochratoxin A; moreover, rubratoxin B decreased fecundity and fertility, especially in diets without yeast, while penicillic acid and oxalic acid had no effects [84].

Dowd [85] evaluated the oral toxicity of several *Penicillium* mycotoxins, alone and in combination, at naturally occurring levels in the fall armyworm (*Spodoptera frugiperda*) and the corn earworm (*Helicoverpa zea*) (Lepidoptera, Noctuidae). As considered alone, ochratoxin A and citrinin were the most toxic products to both species, causing abnormalities in the Malpighian tubules. Their synergistic combination was more toxic to *S. frugiperda*, whereas the combination of ochratoxin A and penicillic acid resulted in greater toxicity to *H. zea*. In another study by the same author based on a series of tremorgenic mycotoxins, penitrem A caused significant mortality to *H. zea* at 25 ppm, while treatment with 2.5 ppm of penitrem A and verruculogen reduced the weights of larvae of both *H. zea* and *S. frugiperda* after 7 days; larvae of *H. zea* were also sensitive at 0.25 ppm of penitrem A. Moreover, paxilline was found to affect larval weight and viability, while paspaline did not cause mortality [86].

The new indole-diterpenoid compound penitrem G was isolated from the mycelium of a strain of *P. crustosum* from corn, along with the already known paspaline and penitrems A–D and F. The latter compounds showed convulsive and insecticidal activities against *O. fasciatus* and the Mediterranean fruit fly (*Ceratitis capitata*: Diptera, Tephritidae); in addition, reductions in the fecundity and fertility were observed in *C. capitata* females treated with a dose of 10 μg. Two functional groups in the penitrem molecule are thought to be implicated in the observed mortality: the chlorine atom seems to be important for the acute mortality, while the epoxy function likely affects the delayed toxicity. In fact, the chlorinated penitrems A, C and F were responsible for the highest acute toxicities, significantly differing from the other non-chlorinated analogues; moreover, penitrems A and F, both possessing an epoxy group, exhibited the highest delayed mortality, while inactivity of penitrem G could depend on the hydroxyl substituent at R_3_ (Figure 1). Paspaline was also inactive [87].

Another series of indole alkaloids were isolated from extracts obtained using various organic solvents from the sclerotioid ascostromata of an isolate of *P. shearii* recovered from savannah soil in the Ivory Coast. Along with the novel compounds shearinines A–C, this series included paxilline and a few of its analogues, namely 21-isopentylpaxilline, 7-hydroxy-13-dehydroxypaxilline, 13-dehydroxypaxilline, 2,18-dioxo-2,18-secopaxilline, and paspalinine. All compounds were responsible for reductions in growth and feeding rates as evaluated in dietary assays against *H. zea* and the dried-fruit beetle (*Carpophilus hemipterus*: Coleoptera, Nitidulidae), with shearinines A–B being the most potent. Shearinine A also exhibited activity in a topical assay against *H. zea*, while shearinine B and paxilline caused significant mortality in a leaf disk assay against *S. frugiperda* [88].

More biosynthetically related indole-diterpenes [89], namely the known janthitrems B–C and the new compounds janthitrems A and D (described as 11,12-epoxyjanthitrems B–C, respectively), were isolated from strains of *P. janthinellum* recovered from pastures in New Zealand. Similar to paxilline used as a positive control, both the new products reduced the weight gain and food consumption of larvae of the porina moth (*Wiseana cervinata*: Lepidoptera, Hepialidae) when added to the diet at the doses of 20 and 50 µg g^−1^, with greater potency shown by janthitrem A [90].

Crude extracts of seven isolates belonging to the species *P. brevicompactum*, *P. citrinum*, and *P. expansum* were assayed for weight reduction and mortality against larvae of the African cotton leafworm (*Spodoptera littoralis*: Lepidoptera, Noctuidae). Five extracts caused significant reductions in weight, and four of them caused significant increases in mortality. A total of 15 secondary metabolites were identified in the extracts, namely brevianamide A, citrinin, cyclopenol, 3,5-dimethyl-6-hydroxyphthalide, 3,5-dimethyl-6-methoxyphthalide, lapidosin, mycophenolic acid, ochratoxin A, patulin, penicillic acid, purpurogenone, rubratoxin B, terrein, viomellein, and xanthocillin. Seven of them were then assayed at 10 ppm against the vinegar fly (*Drosophila melanogaster*: Diptera, Drosophilidae) for feeding inhibition and against *S. littoralis* for feeding inhibition and mortality. Significant effects were observed in all three assays for ochratoxin A, brevianamide A, citrinin, and penicillic acid. Moreover, viomellein significantly reduced feeding in *D. melanogaster* and survival in *S. littoralis*, and cyclopenol significantly inhibited feeding in both insects but not survival of *S. littoralis*, while patulin did not cause significant effects. Finally, high levels of feeding inhibition were obtained for brevianamide A and penicillic acid against *S. littoralis* [91]. Afterward, the same research group reported anti-insect effects against *S. frugiperda* and the tobacco budworm (*Heliothis virescens*: Lepidoptera, Noctuidae) induced by ochratoxin A from *P. verrucosum* and brevianamide A and its photolysis product brevianamide D from *P. viridicatum*. Ochratoxin A and brevianamide A were potent antifeedants against larvae of both species at 1000 ppm, with the latter retaining activity at 100 ppm. Moreover, ochratoxin A caused 100% mortality at 10 ppm against *S. frugiperda*, while brevianamide D was more effective than brevianamide A at reducing pupal weight [92].

#### 3.2.2. Other Products

Assays on *S. frugiperda* were also performed to characterize the insecticidal properties of penifulvin A, a sesquiterpene featuring a [5.5.5.6]dioxafenestrane ring that was purified from an isolate of *P. griseofulvum* from dead wood and was found to cause 74% reduction in the growth rate of larvae when tested at a dietary level of 160 ppm [93]. Moreover, ethyl acetate extracts from fermented rice cultures of two strains colonizing wood-decaying fungi, classified as *P. decaturense* and *P. thiersii*, exhibited potent toxicity against *S. frugiperda* in a dietary assay. Among several novel metabolites possessing anti-insect activity, 15-deoxyoxalicine B and decaturins A–B are characterized by a rare polycyclic structure; the first two compounds were obtained from *P. decaturense*, while decaturin B was produced by *P. thiersii.* Treatment with 15-deoxyoxalicine B at 140 ppm resulted in 23% reduction in growth rate, while decaturin A caused 31% reduction at 100 ppm. Despite the structural similarity (Figure 2), decaturin B showed significantly more potent activity, causing 89% growth reduction at 100 ppm [94]. The known analogues oxalicines A and B were also isolated from these strains, along with three new compounds, namely 15-deoxyoxalicine A and decaturins C–D. Potent anti-insect activity was shown against *S. frugiperda* by oxalicine B, decaturin B, and decaturin D, respectively, causing 62%, 89%, and 77% reductions in growth rate at 100 ppm. Oxalicine A showed comparable activity at this dose and caused 98% growth reduction when tested at a higher level (360 ppm), while decaturin C was less active, causing 28% growth reduction at 80 ppm [95]. Oxalicine B was again purified from butanolic extract of a solid culture of a *Penicillium* strain from soil, provisionally identified as belonging to the subgenus *Furcatum*. It was found to induce 82% mortality against the green peach aphid (*Myzus persicae*: Hemiptera, Aphididae) at 100 ppm, while 32% mortality and weak antifeedant activity resulted at 500 ppm against larvae of the western flower thrips (*Frankliniella occidentalis*: Thysanoptera, Thripidae) [96].

Five alkaloids from cultures of a soil strain of *P. expansum*, communesins A–E, displayed insecticidal activity against third-instar larvae of silkworm (*Bombyx mori*: Lepidoptera, Bombycidae) through oral administration. Communesins B and E respectively exhibited LD_50_ values of 5 and 80 μg g^−1^ in the diet, while communesins A, C, and D were less active [97].

Four new xanthene derivatives, penicixanthenes A–D, were isolated from liquid cultures of a *Penicillium* strain endophytic in the mangrove *Ceriops tagal*. Penicixanthenes B–C inhibited growth of newly hatched larvae of the cotton bollworm (*Helicoverpa armigera*: Lepidoptera, Noctuidae) with IC_50_ values of 100 and 200 μg mL^−1^, respectively; moreover, penicixanthenes A, C and D showed insecticidal activity against freshly hatched larvae of *C. quinquefasciatus* with LC_50_ values of 38.5, 11.6 and 20.5 μg mL^−1^, respectively [98]. The same research group extracted and characterized two new meroterpenoids, penicianstinoids A and B, and eight new isocoumarins, peniciisocoumarins A–H, together with ten known analogues as secondary metabolites of a *Penicillium* isolate from the mangrove *Bruguiera sexangula* var. *rhynchopetala*. Among these compounds, penicianstinoids A–B, peniciisocoumarins A, B, E, F and H, austin, austinol and 1,2-dihydro-7-hydroxydehydroaustin showed growth inhibition activity against newly hatched larvae of *H. armigera* with IC_50_ values ranging from 50 to 200 μg mL^−1^ [99]. Moreover, a new compound obtained from the same strain, penicilactone B, showed insecticidal activity against freshly hatched larvae of *C. quinquefasciatus* with LC_50_ of 78.5 μg mL^−1^ [100]. Two more austin derivatives produced by a soil strain of *P. brasilianum* (MG-11), namely dehydroaustin and acetoxydehydroaustin, displayed insecticidal activity against *A. aegypti*; in particular, the first compound was the most active, with LC_50_ of 2.9 ppm [101].

Significant growth inhibitory properties against newly hatched larvae of *H. armigera*, with IC_50_ values in the range of 50–200 µg mL^−1^, have also been recently reported for secondary metabolites of an isolate of *P. oxalicum* from roots of the mangrove *Lumnitzera littorea*, including four new products, namely the cyclopiane diterpenes conidiogenones J–K, the steroid andrastin H, and the alkaloid (Z)-4-(5-acetoxy-*N*-hydroxy-3-methylpent-2-enamido)butanoate, along with the known compounds demethylincisterol A3, ergosterol, ∆^7^-sitosterol, (−)-β-sitosterol, 7-deacetoxyyanuthone A, and (1*S*,5*R*,6*S*)-5-hydroxy-4-methyl-1-[(2*E*,6*E*)-3,7,11-trimethyl-2,6,10-dodecatrien-1-yl]-7-oxabicyclo [4.1.0]hept-3-en-2-one [102].

A few products of a strain of *P. brevicompactum* were characterized for insecticidal activity against *O. fasciatus*, namely 2-(hept-5-enyl)-3-methyl-4-oxo-6,7,8,8a-tetrahydro-4*H*-pyrrolo [2,1-*b*]-1,3-oxazine [103], along with some compounds of the paraherquamide family and five known diketopiperazines from the culture broth of a strain of *P. cluniae*. Paraherquamide E, with LD_50_ of 0.089 μg, was the most potent product, followed by paraherquamide A; structures of these compounds only differ by the presence of a hydroxyl function in the latter (Figure 2), which was 3.5-fold less active. More in general, comparative structure-activity data indicated that oxidative substitutions in the proline unit of the paraherquamide analogues hinder the insecticidal activity, which conversely is supported by the alkyl substitution [104].

The chloroform and hexane extracts from sclerotioid ascostromata of an isolate of *P. gladioli* (=*Eupenicillium crustaceum*) were found to possess significant anti-insect activity in dietary assays on *H. zea*. The major metabolite responsible for this activity was 10,23-dihydro-24,25-dehydroaflavinine, which was extracted from ascostromata in the amount of 2.8 mg g^−1^. In assays at 3000 ppm, 42% reduction in feeding rate and 79% reduction in weight increment were observed in larvae of *C. hemipterus* and *H. zea*, respectively. New macrophorin-type compounds accounted for the anti-insect activity of ascostromata from another strain of the same species producing no aflavinines, while a strain of *P. egyptiacum* (=*Eupenicillium molle*) was found to synthesize both aflavinines and macrophorins. Moreover, a strain of *P.* (*Eupenicillium*) *reticulisporum* produced the aflavinine analogue 10,23-dihydro-24,25-dehydroaflavinine, along with pyripyropene A [105].

Quinolactacide isolated from solid cultures of a soil strain of *P. citrinum* induced 88% mortality against *M. persicae* at 250 ppm. This compound is characterized by a peculiar structure, in which a quinolone skeleton is conjugated to a bicyclic moiety consisting of a γ-lactam ring and a pyrrole ring. The nitrogen atom of the piperidone ring is not methylated, unlike those at the corresponding position of the related quinolactacins (cf. Section 4.1), indicating that demethylation of the nitrogen atom affects the insecticidal activity [106].

Finally, insecticidal activity against the melon and cotton aphid (*Aphis gossypii*: Hemiptera, Aphididae) at the concentration of 1000 ppm has been reported for penicinoline, a novel pyrrolyl 4-quinolinone alkaloid produced by a mangrove endophytic *Penicillium* strain. In addition to retaining strong activity on this sucking pest, a semi-synthetic lactam obtained through intramolecular dehydration, named penicinotam, also effected total control of the chewing larvae of the diamondback moth (*Plutella xylostella*: Lepidoptera, Plutellidae) at 500 ppm, in addition to displaying some activity on *H. virescens* at 1000 ppm [107].

The implications of secondary metabolites produced by insect-associated *Penicillia* may go beyond a direct anti-insect effect. In fact, it has been observed that patulin produced by *P. urticae* (currently a synonym of *P. griseofulvum*) inhibits conidia germination and growth of *B. bassiana*, possibly resulting in an indirect protective effect for insects when strains producing this metabolite occur in the host microbiome [108]. Indeed, many of the aforementioned compounds and more *Penicillium* secondary metabolites have displayed fungitoxic, antibiotic, and antiviral properties, and they may play protective roles to some extent against entomopathogens [9,11,109,110].

## 4. Effects on the Nervous System

The anti-insect activity of *Penicillium* secondary metabolites derives from their impact on different aspects of insect physiology. The nervous system transmits electric signals, providing a rapid means for sensing the environment and coordinating cellular events and movement [111]. These fundamental functions are targeted by most of the chemicals used for insect pest control [112]. In particular, neurotoxic insecticides act at the neuron–neuron and neuromuscular synaptic contacts, where the propagation of the nervous signals relies on neurotransmitters and their receptors. Ion channels in the postsynaptic membrane are opened after binding of the neurotransmitter to the receptor, and a postsynaptic membrane potential develops. A particular synapse is stimulatory or inhibitory depending on the neurotransmitter released. There is evidence that, in the central nervous system, acetylcholine (ACh) is a synaptic transmitter at stimulatory synapses, while γ-aminobutyric acid (GABA) is an inhibitory transmitter. The chemical transmitters at neuromuscular synapses in insects are l-glutamic acid and possibly l-aspartic acid [113]. Chemical signaling at synapses can be disrupted in many ways by agonists or antagonists of receptors, as well as by inhibitors of the enzymes involved in the turnover of neurotransmitters.

### 4.1. Acetylcholinesterase Inhibitors

Acetylcholinesterase (AChE) is an enzyme involved in the termination of neurotransmission at cholinergic synapses by rapid hydrolysis of acetylcholine and other choline esters, which act as neurotransmitters [114]. Inhibition of AChE is the main mechanism of action of several classes of insecticides of both natural [115] and synthetic origin [116,117]. A variety of assays have been developed to measure AChE activity and the inhibitory effects by fungal extracts and purified compounds [118], including Ellman’s method [119], thin layer chromatography bioautography [120], and a combined liquid chromatography-mass spectrometry modified Ellman’s method [121]. Moreover, some laboratories use the QuantiChrom^®^ assay kit, which is based on an improved Ellman’s method in a 96-well plate reader [122]. Otherwise, assessments can be based on the reduction in *V*_max_ values of the enzyme, as in the case of citreoviridin, which is produced by *P. citreoviride* [123]. Indeed, this kind of bioactivity is commonly investigated in natural product research, considering its multiple applications in agriculture and human medicine [124,125].

In addition to preliminary evidence resulting in studies based on crude culture extracts and their fractions [36,126,127,128,129], more than 60 secondary metabolites purified from *Penicillium* strains from various sources have been characterized for AChE-inhibitory properties (Figure 3). This long list of compounds displaying a notable chemodiversity is to be integrated with huperzine A, the most valuable AChE-inhibitor of natural origin used as a drug for the treatment of Alzheimer’s disease [130]. In fact, this sesquiterpene alkaloid has been reported in several endophytic *Penicillium* strains associated with the firmoss *Huperzia serrata*, from which the product was originally described; in particular, a strain phylogenetically related to *P. citrinum* from Vietnam [131] and strains of *P. polonicum* [132] and *P. griseofulvum* from China [126].

As assessed in terms of IC_50_, the bioactivity of these products ranged from 1 nM for arisugacin A to 280 µM for quinolactacin A1 (Table 2). However, the bioactivity values determined in different laboratories do not always collimate for some compounds (e.g., arisugacins B and D, territrems B and C), implying that results of these assays must be considered provisional until further verification.

First characterized from a strain of *Aspergillus fumigatus* [133], the meroterpenoid pyripyropene A (PPA) has been also reported to be a secondary metabolite of *P. coprobium* [134] and a previously mentioned strain of *P. reticulisporum* [105]. This product represents a fundamental reference for compounds displaying AChE-inhibitory properties, and it has been used as a model for the synthesis of new insecticides [135,136]. More *Penicillium* secondary metabolites present a structural affinity with PPA, such as the arisugacins and the territrems. The previously mentioned oxalicines and decaturins are also biogenetically related to PPA; however, their scaffold, including a pyridinyl-α-pyrone substructure, likely arises from nicotinic acid, acetate, and terpenoid precursors, into which a diterpenoid unit is incorporated instead of the sesquiterpenoid component found in PPA [95].

Other compounds in Figure 2 belong to different classes, also representing models for insights into the structural properties of AChE-inhibitors. This is the case for anthraquinones, such as aloe-emodin and citrorosein [137], xanthones (e.g., pinseline) [138], and chromones (e.g., maritimin and analogues) [139].

In some cases, the experimental findings have provided indications concerning structure–activity relationships, which are valuable in light of their applicative extension. In the case of arisugacins, the enone moiety, the hydroxy group as a substituent of R_5_, and the phenyl group substituents (Figure 3) have been considered to play important roles in AChE inhibition [140]. Likewise, the methyl group on the cycloesadienone of penicitrinone H is essential for inhibitory activity when compared with its B analogue [141], while palitantin was reported to be significantly more potent than its analogue 13-hydroxypalitantin [142]. Some quantitative differences have also been found among stereoisomers, such as (+)- and (-)-penicilliumine [143], quinolactacins A1–A2 [144], and penaloidines A–B, which are unprecedented pyridine alkaloids possessing a tetrahydrofuro[3,2-c][2,7]naphthyridinyl scaffold [145].

In the case of territrem B, an in silico molecular docking study was performed, providing a better understanding of the mechanisms of its inhibitory activity. The compound was determined to tightly bind inside the active pocket of AChE in the same way known for the model compound tacrine. In greater detail, the benzene ring with three methoxy groups is considered to stretch into the catalytic pocket consisting of three amino acid residues (His-440, Phe-330 and Ser-200); four hydrogen bonds are established with two of these residues (His-440 and Ser-200) and Gly-118, while the pyrone ring interacts with Phe-330 [146].

**Table 2 microorganisms-11-01302-t002:** Secondary metabolites of *Penicillium* spp. reported for AChE-inhibitory properties.

Secondary Metabolite	Method	Activity	References
Aloe-emodin	Ellman	42.5 µg/mL (IC_50_)	[147]
Arisugacin A	Ellman	0.001 µM (IC_50_)	[133,148,149]
Arisugacin B	Ellman Ellman	0.0258 µM (IC_50_) 3.03 µM (IC_50_)	[148,149] [150]
Arisugacin C	Ellman Ellman *	2.5 µM (IC_50_) 1.4 µM (IC_50_)	[149] [122]
Arisugacin D	Ellman Ellman	3.5 µM (IC_50_) 53.39 µM (IC_50_)	[149] [150]
Arisugacin F	Ellman	0.37 µM (IC_50_)	[151]
Arisugacin I	Ellman	0.64 µM (IC_50_)	[151]
Arisugacin L	Ellman *	0.191 µM (IC_50_)	[122]
Arisugacin N	Ellman *	3.9 µM (IC_50_)	[122]
Arisugacin O	Ellman *	4.6 µM (IC_50_)	[122]
Arisugacin P	Ellman *	66 µM (IC_50_)	[122]
5-Bromosclerotiorin	Ellman	200 µg/mL (23.87%)	[152]
Citreorosein	Ellman	40.5 µg/mL (IC_50_)	[147]
*Cyclo*-(l-Pro–l-Val)	Marston	10.0 µg (MIR)	[153]
Cyclopenin	Ellman	2.04 µM (IC_50_)	[148]
Dechloroisochromophilone II	Marston	10 ng (MIR)	[154]
3′′-Deoxy-6′-*O*-desmethylcandidusin B	Ellman	7.8 µM (IC_50_)	[155]
6′-*O*-Desmethylcandidusin B	Ellman	5.2 µM (IC_50_)	[155]
Dicitrinin A	Ellman modified	42.0 µM (MIC)	[156]
(3*R*,4*R*)-3,4-Dihydro-4,6-dihydroxy-3-methyl-1-oxo-1*H*-isochromene-5-carboxylic acid	Marston	3.0 µg (MIR)	[157]
(3*R*,4*R*)-4,7-Dihydroxymellein	Marston	10.0 µg (MIR)	[157]
4-(5,7-Dimethoxy-4-oxo-4*H*-chromen-2-yl)butanoic acid	Ellman	93.2 µM (IC_50_)	[158]
3-(5,7-Dimethoxy-4-oxo-4*H*-chromen-2-yl)propanoic acid	Ellman	50.8 µM (IC_50_)	[158]
3-Epiarisugacin E	Ellman	38.23 µM (IC_50_)	[150]
4-Hydroxymellein	Marston	30.0 µg (MIR)	[153]
(*R*)-7-Hydroxymellein	Marston	10.0 µg (MIR)	[157]
13-Hydroxypalitantin	Ellman	12 µM (IC_50_)	[142]
Isochromophilone II	Marston	50 ng (MIR)	[154]
Isochromophilone III	Marston	10 ng (MIR)	[154]
Isochromophilone IV	Marston	100 ng (MIR)	[154]
Isochromophilone VIII	Marston	50 ng (MIR)	[154]
Isocyclocitrinol B	Ellman modified	166.0 µM (MIC)	[156]
Maritimin	Ellman	75.3 µM (IC_50_)	[158]
Ochrephilone	Marston	50 ng (MIR)	[154]
Orcinol	Marston	60.0 µg (MIR)	[153]
(+)-Palitantin	Ellman	0.079 µM (IC_50_)	[142]
Penaloidine A	Ellman	14.85 µM (IC_50_)	[145]
Penaloidine B	Ellman	41.27 uM (IC_50_)	[145]
Penicillar B	Ellman	50 µg/mL (19.5%)	[159]
Penicillar C	Ellman	50 µg/mL (21.3%)	[159]
Penicillic acid	Marston	Not determined	[160]
(+)-Penicilliumine	Ellman	50 µM (32.4%)	[143]
(−)-Penicilliumine	Ellman	50 µM (18.7%)	[143]
Penicinoline	Ellman	87.3 µM (IC_50_)	[161]
Penicinoline E	Ellman	68.5 µM (IC_50_)	[161]
Penicitrinol A	Ellman modified	23.0 µM (MIC)	[156]
Penicitrinone B	Ellman	38.96 µM (IC_50_)	[141]
Penicitrinone H	Ellman	23.62 µM (IC_50_)	[141]
Penicnthene	Ellman	28.03 µM (IC_50_)	[141]
Peniopyranone	Ellman	0.0152 µM (IC_50_)	[162]
Pileotin B	Ellman	13.9 µM (IC_50_)	[163]
Pinselin	Ellman	45.9 µg/mL	[147]
Quinolactacin A1	Ellman	280 µM (IC_50_)	[144]
Quinolactacin A2	Ellman	19.8 µM (IC_50_)	[144]
Sclerotioramine	Ellman	38.7 µM (IC_50_)	[158]
Sorbiterrin A	not reported	25 µg/mL (IC_50_)	[164]
Terreulactone C	Ellman	0.028 µM (IC_50_)	[150]
Territrem A	Ellman	0.11 µM (IC_50_)	[165]
Territrem B	Ellman Ellman Ellman Ellman	0.0076 µM (IC_50_) 0.00703 µM (IC_50_) 0.047 µM (IC_50_) 0.00036 µM (IC_50_)	[133,148] [142] [165] [146]
Territrem C	Ellman Ellman Ellman	0.0068 µM (IC_50_) 0.23 µM (IC_50_) 0.045 µM (IC_50_)	[133,148] [150] [165]
Tetrahydrobisvertinolone	Ellman	50 µg/mL (51.1%)	[166]
Tetrahydrotrichodimer ether	Ellman	50 µg/mL (55.1%)	[166]

MIR: minimum amount required for inhibitory activity; MIC: minimum concentration required for inhibitory activity. * The commercial QuantiChrom^®^ assay kit was employed.

It has been observed that the AChE-inhibitory activity of fungal extracts can be increased by inducing biotic stress following co-inoculation with other fungi [167] or by supplementation of some stimulatory compounds to the growth substrate. For instance, a 100% increase was achieved for an extract from a strain of *P. janthinellum* when it was grown with procainamide [168]. Moreover, cultivation of an endophytic *Penicillium* strain with suberanilohydroxamic acid, a histone deacetylase inhibitor, led to the isolation of two pairs of diterpenic meroterpenoids combining features of PPA and decaturins/oxalicines, namely pyrandecaturins A–B and pileotins A–B, among which the latter pair showed AChE-inhibitory activity [163].

### 4.2. Antagonists of Acetylcholine and γ-Aminobutyric Acid Receptors

In insects, nicotinic acetylcholine receptors (nAChRs) are the most abundant excitatory post-synaptic receptors and represent selective targets for synthetic neurotoxic insecticides, such as neonicotinoids [169]. In addition to direct neurotoxic effects, the latter may affect brain and midgut functions [170], lifespan [171], development [172], reproduction [173], and immunity [174], even at sublethal doses. These findings suggest that the functions of nAChRs and the impact of neurotoxic products targeting these receptors are not limited to the nervous system, as also pointed out by recent studies [175]. In past decades, many compounds that display potent nAChR agonist and antagonist activity have been identified in extracts obtained from fungi [176], some of which are inherent *Penicillium* species/strains (Figure 4).

The neurotoxic meroterpenoid austin and its analogues dehydroaustin and acetoxydehydroaustin, found as products of a previously mentioned strain of *P. brasilianum* (MG-11), induced paralysis in male adult American cockroaches (*Periplaneta americana*: Blattodea, Blattidae) within 1 h after injection. In laboratory assays performed by means of whole-cell patch-clamp electrophysiology, austins proved not to behave as agonists of ACh, GABA, or **l**-glutamate receptors in neurons. When the products were applied before the corresponding ligand, no effect was detected on GABA and **l**-glutamate; conversely, the reduction in ACh- and epibatidine-induced currents indicated an effect as selective antagonists of nAChRs. In particular, dehydroaustin showed the highest blocking potency for nAChRs, differentially attenuating the peak and slowly desensitizing the current amplitude of ACh-induced responses, in an action mode not competitive with ACh [177]. Previous studies on silkworms had shown that, although none of these compounds affected the motility of the caterpillars when tested alone, they enhanced convulsions induced by verruculogen, another tremorgenic product of the same strain [178].

The binding of GABA to insect receptors elicits a rapid, transient opening of anion-selective ion channels, which is generally inhibitory [179]. These receptors are also the targets of several synthetic insecticides, such as fipronil and endosulfan. Alantrypinone, a spiroquinazoline alkaloid isolated from *P. thymicola*, was found to act as a selective antagonist for GABA receptors in houseflies (*Musca domestica*: Diptera, Muscidae). Assays based on a series of synthetic derivatives showed that the amide NHs are important for activity, while removal of indolin-2-one is detrimental [180].

When cultured on okara, which is an insoluble residue of whole soybeans, two soil strains of *P. simplicissimum* produced a series of novel indole alkaloids with a molecular scaffold based on a seven-ring system, named okaramine A–G and J–R, along with some penitrems. Okaramine A–D, G, and Q showed insecticidal activity when added to the diet of third-instar silkworms at the doses of 8, 0.2, 8, 20, 40, and 8 µg g^−1^, respectively. Comparative examination of bioactivities indicated an essential role of the azetidine and azocine moieties (Figure 4); the methoxy group as an R_3_ substituent in okaramine B is also presumed to enhance bioactivity, while the hydroxyl substituent at R_2_ in okaramine D and the *N*-dimethylallyl group in okaramine G reduced effectiveness [181,182,183,184,185,186]. Studies performed by patch-clamp electrophysiology showed that okaramine B induces inward currents that reverse close to the chloride equilibrium potential and are blocked by fipronil. As tested on the GABA-gated chloride channel (GABACl) and the l-glutamate-gated chloride channel (GluCl) in silkworms, okaramine B only activated GluCl [187]. Since GluCl is only found in the nervous systems and muscle cells of invertebrates, okaramines could be regarded as a new lead for the development of safe insect-control products [188].

Two more novel compounds isolated by activity-guided fractionation from okara fermented with a soil isolate (JV-379) of *P. brasilianum*, brasiliamides A–B, were found to induce convulsive effects when added to the diet of silkworms, with ED_50_ values of 300 and 50 μg g^−1^, respectively [189].

## 5. Anti-Juvenile Hormone Activity

Insect metamorphosis is finely regulated by hormones, which are secreted by glands and transported by the circulatory system to other parts of the body, where they evoke physiological responses in target tissues [111]. Juvenile hormone (JH) is a sesquiterpenoid that prevents metamorphosis of insects into the adult stage and is necessary for egg maturation [190]. Any factors inhibiting its synthesis or activity induce precocious interruption of the larval development, determining the formation of abnormal pupae or adults that fail to reproduce [191]. Hence, the discovery of JH inhibitors represents a remarkable target in the search for anti-insect products [192]. So far, several products of *Penicillium* spp. have been reported for their anti-juvenile hormone activity (Figure 4).

Compactin, also known as mevastatin, is best known for its anticholesterolemic properties, characterizing it as the progenitor of a group of blockbuster pharmaceuticals [193]. This compound was reported to be a secondary metabolite of a few *Penicillium* species, such as *P. brevicompactum* and *P. citrinum* [194], and it represents the first product of these fungi to have disclosed potential as an inhibitor of insect growth and development based on its antijuvenile properties. In fact, it was found to competitively inhibit the activity of 3-hydroxy-3-methylglutaryl CoA reductase in homogenates from the corpora allata of the tobacco hornworm (*Manduca sexta*: Lepidoptera, Sphingidae); more specifically, a K_I_ of 0.9 nM was determined for reductase after treatment with the sodium salt of compactin. In intact corpora allata, JH biosynthesis was inhibited by approximately 50% at 10 nM. As an indication of JH deficiency, darkening of the cuticle was observed following injection of compactin into larvae after ecdysis from the third to the fourth instar [195]. In assays performed on the cabbage armyworm (*Mamestra brassicae*: Lepidoptera, Noctuidae), the formation of larval–pupal intermediates was observed after injecting 50 μg of compactin in two or three doses, respectively at 9- or 6-h intervals, beginning 18 h before head capsule slippage in the penultimate instar. Later application 6 h after head capsule slippage only inhibited ommochrome synthesis, an effect that could be prevented by simultaneous application of JH-I. The activity of compactin is reversible, considering that JH biosynthesis inhibition only occurs temporarily due to rapid metabolization [196].

Direct incorporation of compactin into the corpora allata of adult females of *P. americana* inhibited JH-III synthesis. Topical treatment with 100 μg of compactin 12 hr before extirpation of corpora allata resulted in about 50% inhibition of JH-III synthesis when the glands were subsequently cultured in vitro. However, no inhibition of JH synthesis was observed when cockroaches were directly injected [197]. Afterward, similar effects were reported when the product was administered, either free or encapsulated into liposomes, to virgin females of the German cockroach (*Blattella germanica*: Blattodea, Blattellidae). An ID_50_ of 10^−7^ M was estimated for JH synthesis when corpora allata from 6-day-old females were incubated with compactin for 2 h in vitro, an inhibitory effect that persisted for 2 h after treatment. In the case of liposomes containing compactin, significant values of JH inhibition were only induced after prolonged incubation; however, persistence of inhibition was greater in comparison with the non-encapsulated product. Additional experiments showed that compactin moderately inhibited oocyte growth. In fact, administration of the compound in fractionated doses, 3 or 5 × 5 μg, from the third day on the first gonotrophic cycle led to 43% or 28% inhibition, respectively, on day 6. Similar results were obtained for encapsulated compactin, although in this case, a single dose of 5 μg administered on day 3 elicited 26% inhibitory activity. However, neither compactin nor liposomes could inhibit development of the first ootheca in the long term, even if the encapsulated compound caused a significant delay in the gonotrophic cycle when administered at fractionated doses [198].

Two fractions obtained through chromatography of dichloromethane extract from cultures of an isolate of *P. brevicompactum* showed anti-JH activity. One was active when assayed against third-instar nymphs of *O. fasciatus*, whereas the other strongly inhibited JH-III biosynthesis in corpora allata from the migratory locust (*Locusta migratoria*: Orthoptera, Acrididae). Brevioxime was determined as the main component responsible for bioactivity, targeting the final steps of JH-III biosynthesis [199]. Moreover, a new biogenetically related product isolated from the same strain, *N*-(2-methyl-3-oxodecanoyl)-2-pyrroline, also displayed anti-JH activity on *O. fasciatus*, inducing precocious metamorphosis in 70% of nymphs treated with a dose of 10 μg [200]. Subsequently, the same research group reported strong anti-JH activity by the analogue *N*-(2-methyl-3-oxodec-8-enoyl)-2-pyrroline, with ED_50_ at 0.7 μg as determined in assays on newly molted fourth-instar nymphs [109].

In addition to hormone action, insect development and metamorphosis are greatly influenced by the biochemical processes involved in chitin biosynthesis. Therefore, compounds able to interfere at any step in chitin assemblage represent an obvious target for the development of new insecticides. A series of quinazoline-based diketopiperazines were isolated from a marine strain of *P. polonicum* and found to be active against chitinases of the Asian corn borer (*Ostrinia furnacalis*: Lepidoptera, Crambidae), namely the three new polonimides A–C and the known aurantiomides A–C and anacine. Particularly, these compounds displayed strong activity against GH18 chitinase, along with weak inhibitory effects toward GH20 β-*N*-acetyl-d-hexosaminidase at a concentration of 10.0 μM, with inhibition rates ranging between 79.1–95.4% and 0.7–10.3%, respectively. The authors of this study also provided insights into the binding mode of these compounds based on molecular docking, obtaining some clues about the structure–activity relationships. In greater detail, the methoxy group in polonimide A could weaken the inhibitory activity compared to aurantiomide C; moreover, the double bond in the Z configuration improved the activity of aurantiomide C compared to polonimide B, indicating a direct influence of the geometry of the double bond [201].

## 6. Effects on Immune Response

Insect immune responses represent an emerging target for developing new control strategies based on reducing immunocompentence to enhance the action of entomopathogens used as bioinsecticides, such as *Bacillus thuringiensis* and *B. bassiana* [202]. When an entomopathogen overcomes the physical barriers of the insect host, the cellular immune response is rapidly activated through the processes of phagocytosis, nodulation, and encapsulation by hemocytes. These reactions are usually accompanied by the release of coagulating and melanizing agents, which promote wound closure and antibiosis against the invading organisms [113]. The coevolutionary arms race between fungal entomopathogens and their hosts has led to the diversification of sophisticated strategies to counter insect immune defenses [203]. Secondary metabolites are part of such strategies, with their effects on cellular and humoral responses by the host during infection [204,205].

Being involved in melanization and wound healing, phenoloxidase (PO) plays a fundamental role in the defense reactions of insects against pathogens and parasites [206]. In fact, this enzyme generates quinone compounds and other reactive intermediates, which are effective in immobilizing and killing the invaders [207]. One of the best-known examples of a PO inhibitor is represented by kojic acid (Figure 4), which has been reported as a secondary metabolite in many fungal genera, including *Penicillium* [208]. This compound induced 50% inhibition of activity of hemolymph serum PO from larvae of *S. littoralis* at a concentration of 135 μM [209]. Moreover, at 10^−1^ M, kojic acid inhibited PO activity in the cuticle and hemolymph of *S. frugiperda* by 80%, and significant inhibition (35–40%) was still detectable at 10^−4^ M. Inhibition of PO activity by kojic acid was also observed in *H. zea*, *L. serricorne*, and the Freeman sap beetle (*Carpophilus freemani*: Coleoptera, Nitidulidae), while it was much less effective against greenhouse whiteflies (*Trialeurodes vaporariorum*: Heteroptera, Aleyrodidae) [210].

Kojic acid also proved to disrupt the development of the pumpkin fruit fly (*Zeugodacus tau*: Diptera, Tephritidae) by acting as an effective competitive inhibitor of PO. When larvae were fed a diet with 1.66% kojic acid added, the levels of *ZtPPO1* transcripts, which are highly expressed during larval–prepupal transition and in the hemolymph, significantly increased by 2.79 and 3.39 fold in the whole larvae and cuticle, respectively, while the corresponding PO activity was significantly reduced; in addition, the larval and pupal instar durations were significantly prolonged, the pupal weights were lower, and abnormal phenotypes originated [211].

More *Penicillium* secondary metabolites would deserve circumstantial investigations concerning their inhibitory properties against PO and other enzymes involved in the immune response, as well as the plasma hemolymph components. Recently, citrinin from *P. aurantiogriseum* has been reported to induce effects on PO activity, hemocytes, and total hemolymph protein in fifth-instar nymphs of the desert locust (*Schistocerca gregaria*: Orthoptera, Acrididae) up to 9 days post-application. The treated nymphs revealed fluctuations in the mean plasmatocyte, lymphocyte, granulocyte, and total hemocyte counts, along with alteration of PO titers at all intervals after infection [23]. Other products could have more general effects on the immune system. This is the case with phoenicin, a dimer deriving from two 2-hydroxy-6-methyl-benzoquinones, which is produced by several *Penicillium* species in the sections *Charlesia, Citrina* and *Exilicaulis* [212,213]; in fact, it is structurally related to oosporein, a bibenzoquinone product typical of *Beauveria* spp., which is known to possess immunosuppressive properties toward insects [204,214].

## 7. Behavioral Effects

Many smart approaches to pest control address the manipulation of insect behavior through the use of semiochemicals. These signaling molecules, which are emitted from host plants or other potential partners, and received by olfactory organs, may lure insects to a bait (‘pull’), or repel them from potential host plants or animals (‘push’). Within this framework, many natural compounds, such as pheromones, plant volatiles, essential oils, and proteins, can be used as insect attractants or repellents [215].

The azadirachtins, tetranortriterpene limonoids typical of the neem tree (*Azadirachta indica*) and related species among the Meliaceae, are a well-known example of a natural insecticide, which basically acts as an antifeedant and growth disruptor. Interestingly, a neem endophytic strain of *P. parvum* was reported to produce azadirachtins A–B [216]. Likewise, the pyranone compounds phomopsolides A–B, known as feeding deterrents for elm bark beetles (*Scolytus* spp.: Coleoptera, Scolytinae), have been extracted from liquid cultures of an unidentified *Penicillium* strain endophytic in the Pacific yew (*Taxus brevifolia*) [217] and a strain of *P. clavigerum* associated with the green alga *Chlorella vulgaris* [218].

Xanthomegnin and the already mentioned viomellein are bis-naphthoquinones known as products of several *Penicillium* species [219]. The deterrent effects of these mycotoxins were tested in a food choice experiment by adding them to bakery yeast offered to the springtail *Folsomia candida* (Collembola, Isotomidae). At the level of 10 mg g^−1^, the springtails avoided yeast containing either viomellein or xanthomegnin, and no insects were feeding after 20 min of exposure; at a lower level (2 mg g^−1^), the deterrent effects were less noticeable, even if still significant [220].

Known to play a role in inter- and intraspecific communication in many organisms, including insects, volatile organic compounds (VOCs) already have practical application as attractants/deterrents with the aim of disrupting this remote signaling. Products in this category have also been reported from *Penicillium* species associated with plants and insects [44]. This is the case of an isolate of *P. expansum* from the frass and feces of the pine weevil (*Hylobius abietis*: Coleoptera, Curculionidae). VOCs produced by this strain in cultures on sterilized pine frass medium were collected by solid phase micro-extraction; styrene and 3-methylanisole were found to be the major products through GC-MS analysis. In particular, large quantities of styrene were produced when the fungus was cultured on grated pine bark with yeast extract. In a multi-choice arena test, styrene significantly reduced the attraction of male and female weevils to pieces of pine twigs, whereas 3-methylanisole only reduced the attraction of males [48].

VOCs emitted by strains of several *Penicillium* species, namely *P. expansum*, *P. solitum*, *P. crustosum*, *P. polonicum* and *P. maximae*, caused repellent and toxic effects on *Drosophila* larvae and adults [221]. Low concentrations of the vapor form of several eight carbon compounds, including oxylipins and 1-octen-3-ol, were found to be toxic to *D. melanogaster* in previous studies [222,223,224]. In particular, 1-octen-3-ol selectively induced behavioral alterations upon affection of dopaminergic neurons and inflammatory responses in hemocytes [225,226]. Moreover, geosmin, known as a volatile product of several *Penicillium* spp. [219], was determined to be an ecological stimulus that alerts *Drosophila* flies to the presence of harmful microbes. In fact, this compound activates a single class of sensory neurons expressing the olfactory receptor Or56a. These neurons connect to projection neurons that respond exclusively to geosmin, with an aversive effect that overcomes inputs from other olfactory pathways and inhibits positive chemotaxis, feeding, and oviposition. Hence, the geosmin detection system enables flies to identify unsuitable feeding and breeding sites [227]. Conversely, rather than being aversive, geosmin was found to mediate egg-laying site selection in *A. aegypti;* female mosquitoes likely associate geosmin with the availability of microbes representing the food sources of larvae, supporting possible use of this compound as attractant in baits [228].

Gravid females of the oriental fruit fly (*Bactrocera dorsalis*: Diptera, Tephritidae) were found to be attracted by VOCs emitted by a strain of *P. citrinum*. After finding the fungus in the intestinal tracts throughout all larval stages, a mutualistic association between the two organisms based on nutritional supplementation was postulated, which could be considered a possible target in control programs [229]. Likewise, three *Penicillium* species, namely *P. citrinum*, *P. sumatrense*, and *P. digitatum*, were found to influence the oviposition selection and behavior of the yellow peach moth (*Conogethes punctiferalis*: Lepidoptera, Crambidae). When comparing infected with non-infected, mechanically damaged apples and strains of the three *Penicillium* species cultured on potato dextrose agar, both the oviposition selection and four-arm olfactometer experiments showed that mated moth females preferred apples infected by the fungi. Further GC-MS analyses of VOCs showed that the absolute contents of ethyl hexanoate and (*Z*,*E*)-α-farnesene in the presence of these fungal isolates were higher than those in non-infected apples; a total of 16 novel VOCs were detected in apples infected by fungi, demonstrating a change in the components and proportions of apple VOCs. This finding paves the way for the development of new field trapping strategies to be adopted in the management of *C. punctiferalis* [230]. In another study by the same research group, a repellent effect on mated females of this moth, which is also a pest of maize, was observed by VOCs produced by *P. oxalicum*. Since the latter is known as a pathogen causing maize ear rot, possible effects on the phytosanitary conditions of this crop may be derived as a consequence of the natural interaction between these two organisms [231].

## 8. Conclusions

The examination of the pertinent literature points out a widespread occurrence of *Penicillium* species/strains in association with insects from most orders, as well as various ecological contexts and geographical areas. Although this finding is in line with the known ubiquity of these micromycetes, the aptitude of these fungi as producers of secondary metabolites disclosing anti-insect properties stimulates a more accurate consideration of their ecological role and the real effects of these symbiotic interactions. In particular, the diffuse endophytic association of *Penicillia* with plants has increasingly emerged in past decades [11,232]; it is indicative of a possible direct impact on herbivorous insects, which deserves to be thoroughly investigated.

More than 150 compounds belonging to several chemical classes have been reported from *Penicillium* spp. and considered in our overview, confirming chemodiversity as a notable feature of these fungi. However, in most instances, the assessment of their effects has been undertaken following a general intent to detect any bioactivity or the propensity of single laboratories for investigations concerning a specific mode of action. Indeed, most of the available data are partial or fragmentary and do not offer an exhaustive appreciation of the real potential of the compounds and the producing strains in insect pest management. In this respect, the establishment of definite protocols to be followed in the aim of providing a more circumstantial account of the anti-insect properties for both new and old products could result in substantial progress in view of applicative perspectives. Following the example of pyripyropene A, the expectation is great that multiple products could be selected as new leads for the development of ecofriendly insecticides.

## Figures and Tables

**Figure 1 microorganisms-11-01302-f001:**
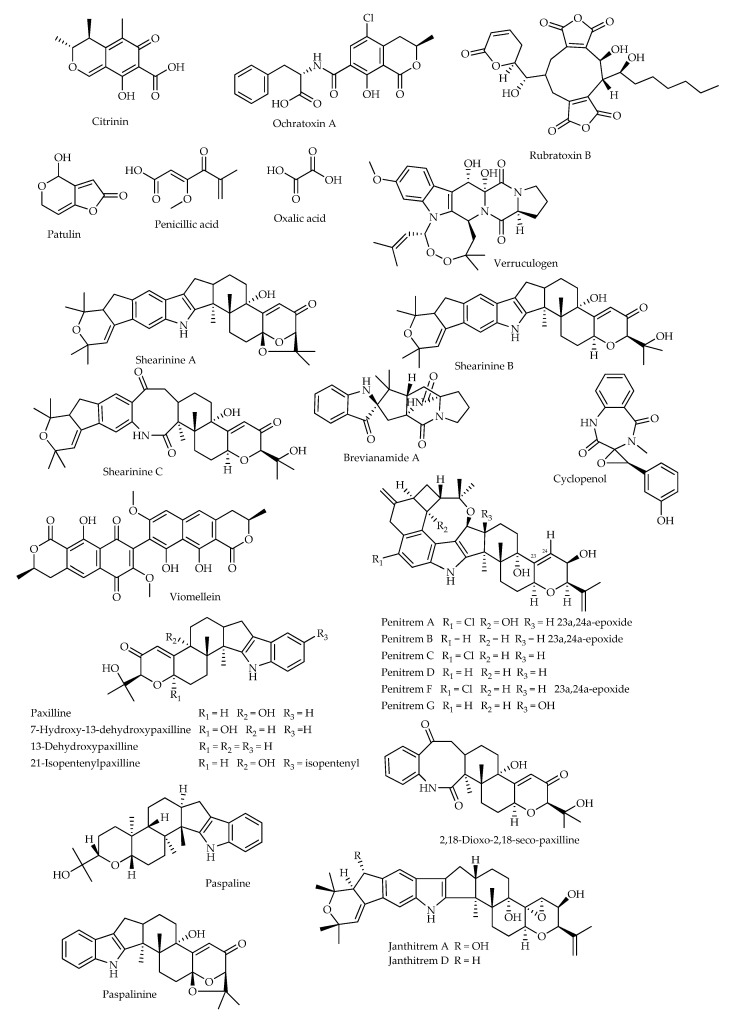
Chemical structures of mycotoxins from *Penicillium* spp. with effects on insect viability and development.

**Figure 2 microorganisms-11-01302-f002:**
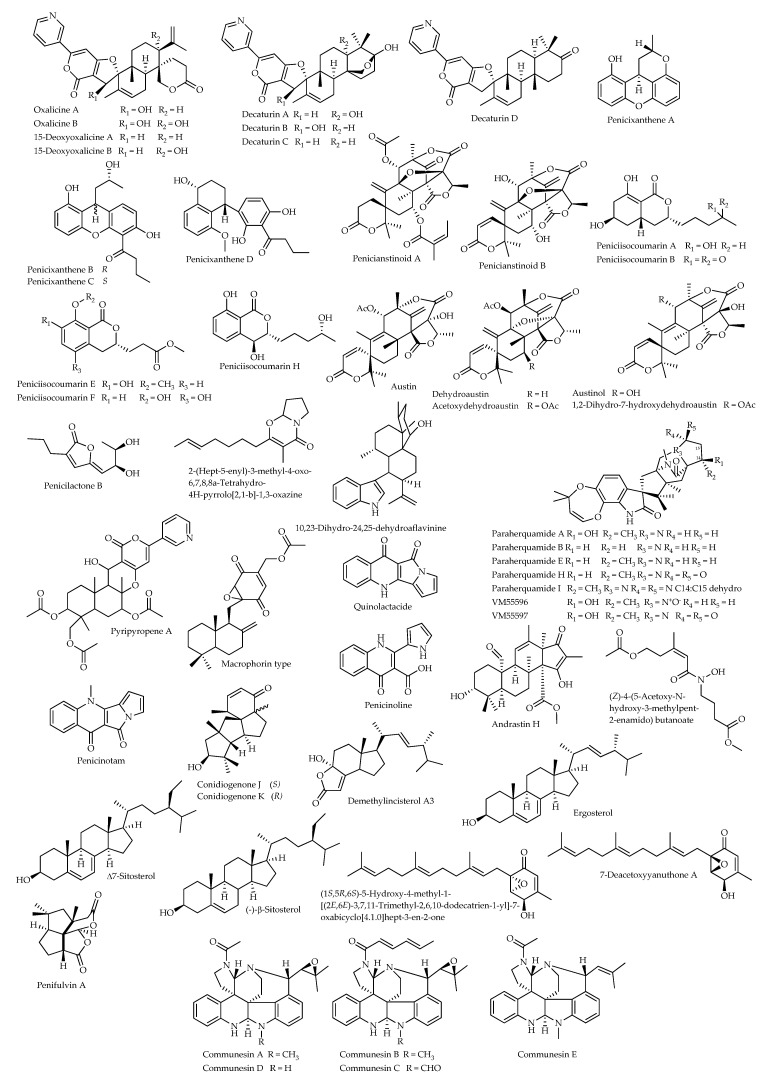
Structural diversity of secondary metabolites from *Penicillium* spp. displaying anti-insect activities.

**Figure 3 microorganisms-11-01302-f003:**
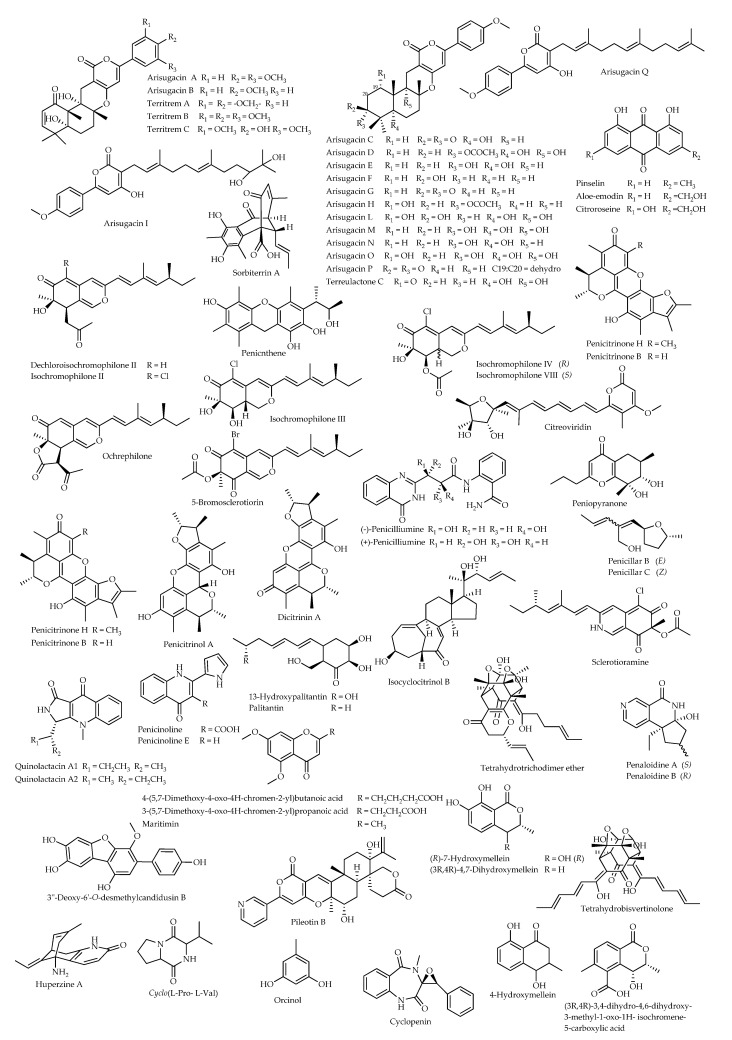
Chemical structures of compounds from *Penicillium* spp. with AChE-inhibitory properties.

**Figure 4 microorganisms-11-01302-f004:**
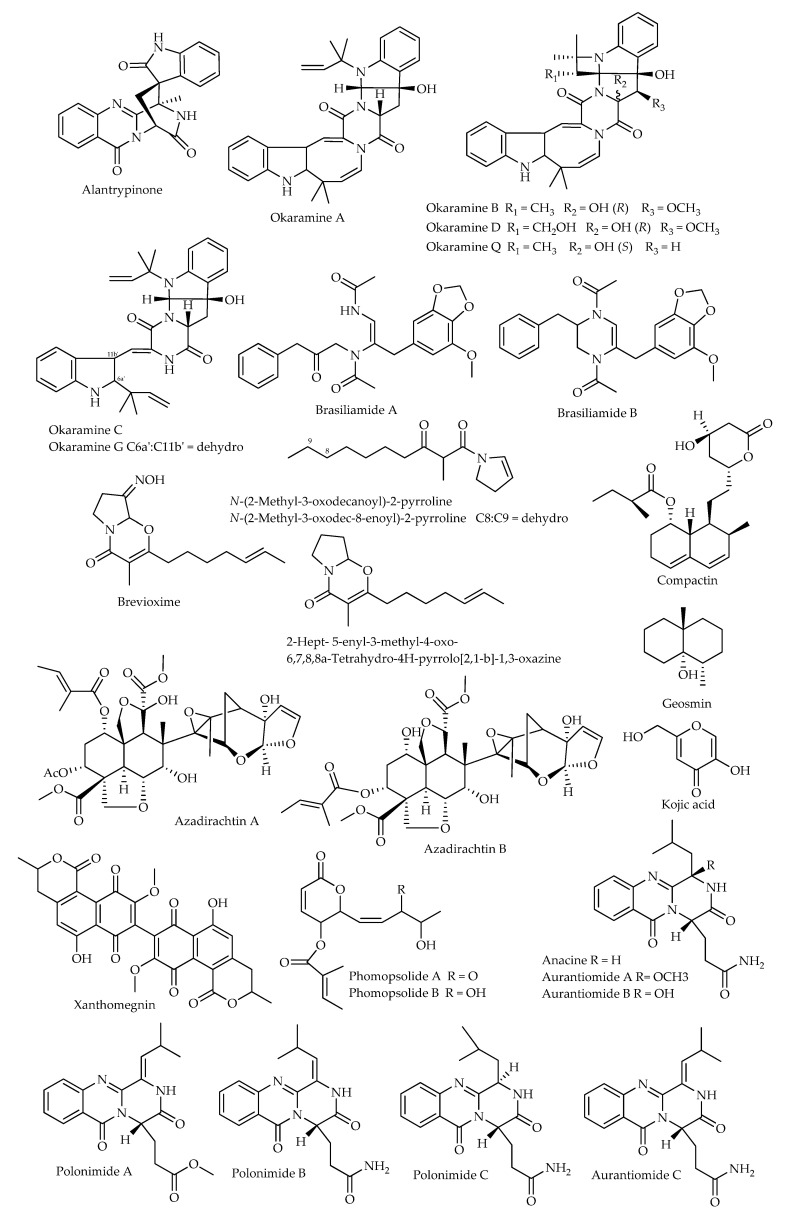
Chemical structures of compounds of *Penicillium* spp. displaying effects as antagonists of ACh receptors regarding juvenile hormone activity and insects’ immune responses and behaviors.

**Table 1 microorganisms-11-01302-t001:** *Penicillium* species reported in association with insects.

Species	Host	Country	Source	References
*P. adametzioides*	unspecified insect	Brazil	dead or live insect	[19]
*P. atrofulvum*	*Triplectides* sp.	Brazil	gut	[20]
*P. aurantiocandidum*	*Vespula vulgaris*	California (USA)	gut of larva	[21]
*P. aurantiogriseum*	*Ostrinia nubilalis*	Iowa (USA)	dead adult and larva	[22]
	*Schistocerca gregaria*	Egypt	dead adult	[23]
	*Spodoptera littoralis*	Egypt	eggs	[24]
	*Triatoma brasiliensis*	Brazil	digestive tract	[25]
*P. brasilianum*	unidentified insect	South Korea	dead insect	[26]
*P. brevicompactum*	*Apis mellifera*	Sweden	midgut	[27]
	*Bradysia agrestis*	South Korea	gut	[28]
	*Malacosoma neustria*	Germany	diseased larvae	[29]
*P. brocae*	*Hypothenemus hempei*	Mexico	cuticle and frass of females	[30]
*P. cairnsense*	*Triplectides* sp.	Brazil	gut	[20]
*P. camponotum*	*Camponotus herculeanus*	Germany	buccal cavities	[31]
	*Camponotus pennsylvanicus*	Canada		
*P. canescens*	*Aedes* sp.	Brazil	adult	[32]
*P. caseifulvum*	*Triplectides* sp.	Brazil	gut	[20]
*P. chermesinum*	*Vespula pennsylvanica*	California (USA)	gut of larva	[21]
*P. chrysogenum*	*A. mellifera*	Michigan (USA)	dead adult	[33]
		Arizona (USA)	gut	[34,35]
	*Culex nigripalpus*	Brazil	diseased adult	[36]
	*Culex* sp.	Brazil	adult	[32]
	*Cydia ulicetana*	New Zealand	exoskeleton of live adult	[37]
	*S. littoralis*	Egypt	pupae and adults	[24]
	unspecified insect	Brazil	dead or live insect	[19]
*P. cinnamopurpureum*	*Triatoma pseudomaculata*	Brazil	digestive tract	[25]
*P. citreonigrum*	*Sericothrips staphylinus*	New Zealand	exoskeleton of live adult	[37]
	*T. brasiliensis*	Brazil	digestive tract	[25]
*P. citrinum*	*Aedes aegypti*	Australia	eggs	[38]
	*Antheraea mylitta*	India	diseased larvae	[39]
	*A. mellifera*	Arizona (USA)	gut	[35]
	*B. agrestis*	South Korea	gut	[28]
	*Culex quinquefasciatus*	Thailand	dead adult	[40]
	*Culex* sp.	Brazil	adult	[32]
	*H. hempei*	Mexico	cuticle and gut of females	[30]
	*Nomia melanderi*	western USA	diseased larva	[41]
	*Parastrongylus megistus*	Brazil	digestive tract	[42]
	*Pteroptyx bearni*	Sabah (Malaysia)	eggs	[43]
	*T. brasiliensis*, *T. infestans*	Brazil	digestive tract	[25]
	*Triplectides* sp.	Brazil	gut	[20]
	unspecified insect	Brazil	internal mycobiota	[19]
	*V. vulgaris*	California (USA)	larva	[21]
*P. coffeae*	*B. agrestis*	South Korea	gut	[28]
*P. commune*	*A. mellifera*	Michigan (USA)	dead adults	[33]
	*Hylobius abietis*	Sweden	frass	[44]
*P. corylophilum*	*Anopheles darlingi*, *Culex declarator*, *C. nigripalpus*, *C. saltanensis*, *Mansonia titilans*	Brazil	larvae, adults	[7,32,36]
	*A. mellifera*	Michigan (USA)	dead adults	[33]
		Arizona (USA)	gut	[35]
	*Musca domestica*	Brazil	diseased adult/larva	[36]
	*P. megistus*	Brazil	digestive tract	[42]
	*T. infestans*, *T. sordida*, *T. vitticeps*, *T. brasiliensis*, *T. pseudomaculata*	Brazil	digestive tract	[25,36]
	*V. pennsylvanica*	California (USA)	adult	[21]
*P. costaricense*	*Rothschildia lebeau*	Costa Rica	gut	[31]
*P. crustosum*	*H. hempei*	Mexico	cuticle, frass, gut of female	[30]
*P. cyclopium*	*A. mellifera*	Michigan (USA)	dead adults	[33]
		Arizona (USA)	gut	[35]
	*Ostrinia nubilalis*	Iowa (USA)	dead adult and larva	[22]
*P. decumbens*	*Aedes* sp., *Anopheles* sp., *Culex* sp., *Mansonia* sp.	Brazil	adult	[32]
	*Helicoverpa zea*	Iowa (USA)	dead larva and pupa	[22]
	*Ostrinia nubilalis*	Iowa (USA)	dead larva	[22]
	*P. megistus*	Brazil	digestive tract	[42]
	*T. brasiliensis*	Brazil	digestive tract	[25]
	*V. pennsylvanica*	California (USA)	gut of larva	[21]
*P. euglaucum*	*A. mellifera*	Italy	dead/live adults/larvae	[45]
	*N. melanderi*	northwestern USA	diseased larvae	[46]
*P. excelsum*	bees, ants	Brazil	adults	[47]
*P. expansum*	*A. mellifera*	Michigan (USA)	dead adults	[33]
	*Anopheles* sp., *Culex* sp.	Brazil	adult	[32]
	*H. abietis*	Sweden	frass	[44,48]
	*T. brasiliensis*	Brazil	digestive tract	[25]
*P. exsudans*	*Triplectides* sp.	Brazil	gut	[20]
*P. fellutanum*	*Aedes scapularis*	Brazil	diseased adult	[36]
	*C. quinquefasciatus*	Brazil	larvae, adults	[7,32]
	*P. megistus*	Brazil	digestive tract	[42]
	*T. brasiliensis*, *T. infestans*	Brazil	digestive tract	[25]
*P. freii*	unidentified Pyralidae	Lebanon	dead moth	[49]
*P. fundyense*	*C. pennsylvanicus*	Canada	buccal cavities	[31]
*P. glabrum*	*A. mellifera*	Arizona (USA)	gut	[34,35]
	*T. brasiliensis*	Brazil	digestive tract	[25]
*P. gladioli*	*B. agrestis*	South Korea	gut	[28]
*P. griseofulvum*	*A. mellifera*	Arizona (USA)	gut	[34]
	*T. infestans*	Brazil	digestive tract	[25]
		Argentina	digestive tract	[50]
*P. guanacastense*	*Eutelia* sp.	Costa Rica	gut	[51]
*P. herquei*	*Euops chinensis*	China	mycangia	[15]
*P. infrabuccatum*	*C. pennsylvanicus*	Canada	buccal cavities	[31]
*P. janthinellum*	*Aedes fluvialitis*, *A. darlingi*, *C. quinquefasciatus*	Brazil	larvae, adults	[7,32]
	*Anopheles* sp., *C. nigripalpus*	Brazil	diseased adult	[36]
	*P. megistus*	Brazil	digestive tract	[42]
	*T. infestans*, *T. brasiliensis*	Brazil	digestive tract	[25]
*P. lanosum*	*V. pennsylvanica*	California (USA)	gut of larva	[21]
*P. lividum*	*H. abietis*	Sweden	frass	[44]
*P. mallochii*	*Citheronia lobesis*, *R. lebeau*	Costa Rica	gut	[51]
	*Triplectides* sp.	Brazil	gut	[20]
	unspecified insect	Brazil	dead or live insect	[19]
*P. maximae*	*Triplectides* sp.	Brazil	gut	[20]
*P. miczynskii*	*T. infestans*	Brazil	digestive tract	[25]
*P. ochrochloron*	*A. mellifera*	Arizona (USA)	gut	[34]
*P. olsonii*	*H. hempei*	Mexico	cuticle and gut of females	[30]
*P. oxalicum*	*Acrida bicolor*	China	gut	[52]
	*Bemisia tabaci*	India	mummified adult	[53]
	*Mansonia* sp.	Brazil	adult	[32]
*P. palitans*	*A. mellifera*	Michigan (USA)	live adults	[33]
*P. paxilli*	*Triplectides* sp.	Brazil	gut	[20]
*P. phoeniceum*	*V. pennsylvanica*	California (USA)	gut of larva	[21]
*P. polonicum*	*Culex* sp.	Lebanon	dead adult	[49]
	*Muljarus japonicus*	South Korea	dead adult	[54]
*P. restrictum*	*A. aegypti*	Australia	eggs	[38]
	*V. pennsylvanica*	California (USA)	gut of larva	[21]
*P. rolfsii*	*Triplectides* sp.	Brazil	gut	[20]
*P. roseopurpureum*	*A. aegypti*	Australia	eggs	[38]
*P. rubens*	*Triplectides* sp.	Brazil	gut	[20]
*P. rubidurum*	*B. agrestis*	South Korea	gut	[28]
*P. simplicissimum*	*A. aegypti*	Australia	eggs	[38]
*P. solitum*	*H. abietis*	Sweden	frass	[44]
*P. soppii*	*Formica polyctena*	Poland	worker	[55]
*Penicillium* sp.	*Acromyrmex balzani*, *A. niger*, *A. rugosus*, *A. subterraneus*	Brazil	workers	[56]
	*A. aegypti*	Australia	eggs	[38]
	*A. darlingi*	Brazil	diseased adult	[36]
	*A. mellifera*	Costa Rica	diseased larvae	[57]
	*Atta colombica*	Panama	queen cuticle	[58]
	*Bombus* sp.	Denmark	dead adult	[59]
	*Carpophilus dimidiatus*, *Catarthus quadricollis*, *Cryptolestes ferrugineus*, *Gnathocerus cornutus*, *Palorus subdepressus*, *Prostephanus truncatus*, *Sitophilus zeamais*, *Tribolium castaneum*	Benin	adults	[60]
	*C. quinquefasciatus*	Thailand	dead adult	[40]
	*Diaphania* (*Margaronia*) *pyloalis*	Japan	larva	[61]
	*Euops lespedezae*	Japan	mycangia	[62]
	*F. polyctena*	Poland	workers	[55]
	*Halictus rubicundus*	India	frass	[63]
	*Lasioglossum zephyrum*	India	dead larvae	[63]
	*Lixus impressiventris*	South Korea	dead insect	[54]
	*M. domestica*	Iran	adults	[64]
	*M. domestica*	Brazil	diseased adult/larva	[36]
	*Periplaneta americana*	Sumatra (Indonesia)	adult	[65]
	*Platypus cylindrus*	Portugal	exoskeleton, gut, mycangia	[66]
	*Reticulitermes flavipes*	Ontario (Canada)	live termites	[67]
	*Scaptocoris carvalhoi*	Brazil	adult or nymph	[68]
	*T. brasiliensis*, *T. pseudomaculata*, *T. vitticeps*	Brazil	digestive tract	[25]
	*T. infestans*	Argentina	digestive tract	[50]
	*Tribolium castaneum*	India	adults	[69]
	*Zonocerus variegatus*	Nigeria	dead adult	[70]
*P. spinulosum*	*H. abietis*	Sweden	feces and frass	[44]
	*T. brasiliensis*, *T. pseudomaculata*	Brazil	digestive tract	[25]
*P. steckii*	*Meloe proscarabaeus*	South Korea	dead adult	[54]
	*P. megistus*	Brazil	digestive tract	[42]
	*T. sordida*	Brazil	digestive tract	[25]
	*V. vulgaris*	California (USA)	larva	[21]
*P. vancouverense*	*Triplectides* sp.	Brazil	gut	[20]
*P. viridicatum*	*A. darlingi*	Brazil	larvae, adults	[7,32]
	*T. brasiliensis*	Brazil	digestive tract	[25]
*P. waksmanii*	*Aedes* sp., *Anopheles* sp., *C. quinquefasciatus*, *M. titilans*	Brazil	larvae, adults	[7,32,71]
	*P. megistus*	Brazil	digestive tract	[42]
	*T. brasiliensis*, *T. pseudomaculata*, *T. vitticeps*	Brazil	digestive tract	[25]

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
