# Peer review of "Anti-Insect Properties of Penicillium Secondary Metabolites"

_microorganisms, 2023, doi:10.3390/microorganisms11051302_

Round 1

Reviewer 1 Report

I would like to express my delight of masterpiece work Antiinsectan properties of Penicillium secondary metabolites. This review work was made on high science level. It was a great pleaser to read it. Thank you for scientific regard to material. This work contains the good References with both modern works and basic old investigations. There are several misprints in work:

Line 160: should be “mycotoxin” instead of “mvcotoxin”;

Line 998: should be “wasp”instead of “wqasp”.

Author Response

Thank you for your suggestions, we changed the text accordingly.

Reviewer 2 Report

The review "Antinsectan Properties of Penicillium Secondary Metabolites" by Nicoletti and coworkers explores secondary metabolites derived from Penicillium species with diverse activities against insects and their potential applications. The review is well-written and delves into an interesting subject. I have only a few suggestions.

In general, some unit conversions would be appreciated, especially when dealing with ppm.

Line 152: P-450

Line 170-172: Which concentrations were evaluated?

Line 307: "Spanish research group" - I don't think there is a need to describe the nationality of the research groups.

It would be helpful to link compounds with numbers for easier location.

Is cyclophostin from fungi? (Table 2)

I would not describe epigenetic inhibitors as "stimulatory compounds."

Author Response

Thank you for your suggestions and appreciation. Here's a point by point response to your comments:

The review "Antinsectan Properties of Penicillium Secondary Metabolites" by Nicoletti and coworkers explores secondary metabolites derived from Penicillium species with diverse activities against insects and their potential applications. The review is well-written and delves into an interesting subject. I have only a few suggestions.

In general, some unit conversions would be appreciated, especially when dealing with ppm.

Line 152: P-450

  • We feel “P-450” spelling is correct.

Line 170-172: Which concentrations were evaluated?

  • Done

Line 307: "Spanish research group" - I don't think there is a need to describe the nationality of the research groups.

  • Done

    It would be helpful to link compounds with numbers for easier location.

  • We feel that adding numbers to compounds at this stage will twist the whole manuscript and decrease readability.

Is cyclophostin from fungi? (Table 2)

  • Thank you for your suggestion. Indeed, we erroneously included cyclophostin, which is derived from bacteria and used by the cited authors as a comparison compound. However, we removed it from the text. We apologize for such mistake.

I would not describe epigenetic inhibitors as "stimulatory compounds."

  • At line 447-449 “stimulatory compunds” is not referred to AChE (acetylcholinesterase ) inhibitors, but to other compounds added to the culture substrate which may stimulate the biosynthesis of acetylcholinesterase inhibitors by fungi. So, we did not make any change.

Reviewer 3 Report

This manuscript provided a good review on secondary metabolites of Penicillium. spp that have displayed antiinsectan effects and potentials to be promissing pestcides. The results and figures were presented clearly, and writing is also appropriate. I think it is a good review.

Author Response

Thank you for your appreciation.